

# Transcriptomics reveals the effects of NTRK1 on endoplasmic reticulum stress response-associated genes in human neuronal cell lines

Bo Jiao[1,*], Mi Zhang[1,2,*], Caixia Zhang[1], Xueqin Cao[1], Baowen Liu[1], Ningbo Li[1], Jiaoli Sun[1] and Xianwei Zhang[1]

[1] Department of Anesthesiology, Tongji Hospital of Tongji Medical College, Huazhong University of Science and Technology, Wuhan, Hubei Province, China
[2] Department of Anesthesiology, Zhongnan Hospital, Wuhan University, Wuhan, Hubei Province, China
* These authors contributed equally to this work.

## ABSTRACT

**Background:** *NTRK1* gene, encoding TrkA, is essential for the nervous system and drives a variety of biological processes, including pain. Given the unsatisfied analgesic effects of some new drugs targeting *NTRK1* in clinic, a deeper understanding for the mechanism of *NTRK1* in neurons is crucial.

**Methods:** We assessed the transcriptional responses in SH-SY5Y cells with *NTRK1* overexpression using bioinformatics analysis. GO and KEGG analyses were performed, PPI networks were constructed, and the functional modules and top 10 genes were screened. Subsequently, hub genes were validated using RT-qPCR.

**Results:** A total of 419 DEGs were identified, including 193 upregulated and 226 downregulated genes. GO showed that upregulated genes were mainly enriched in response to endoplasmic reticulum (ER) stress, protein folding in ER, *etc.*, and downregulated genes were highly enriched in a series of cellular parts and cellular processes. KEGG showed DEGs were enriched in protein processing in ER and pathways associated with cell proliferation and migration. The finest module was dramatically enriched in the ER stress response-related biological process.

The verified seven hub genes consisted of five upregulated genes (COL1A1, P4HB, HSPA5, THBS1, and XBP1) and two downregulated genes (CCND1 and COL3A1), and almost all were correlated with response to ER stress.

**Conclusion:** Our data demonstrated that *NTRK1* significantly influenced the gene transcription of ER stress response in SH-SY5Y cells. It indicated that ER stress response could contribute to various functions of *NTRK1*-dependent neurons, and therefore, ER stress response-associated genes need further study for neurological dysfunction implicated in *NTRK1*.

Corresponding author
Xianwei Zhang, ourpain@163.com

## INTRODUCTION

Neurotrophic tyrosine kinase receptor type 1 (*NTRK1*) gene encodes tropomyosin receptor kinase A (TrkA), a member of the neurotrophic tyrosine receptor kinase (NTRK) family, and is expressed in various tissues and organs, such as the nervous system, and it is located in the cytosol and vesicles in cells. TrkA, acted by nerve growth factor (NGF), plays a major part in embryonic neuronal development, and in several biological processes in adults, including pain, homeostasis, inflammation, and emotions and feelings (*Indo, 2018*). Encouragingly, the *NTRK* gene has been precisely targeted for anticancer treatment recently (*Klink et al., 2022*). These data suggested the powerful enchantment of the *NTRK1* gene. Although there are several mechanisms that were identified previously, the underlying mechanism of the *NTRK1* is still not fully understood.

Congenital insensitivity to pain with anhidrosis (CIPA) is caused by loss-of-function mutations in the *NTRK1* gene. It is characterized by insensitivity to pain, anhidrosis, mental retardation, and characteristic behaviors (*Li et al., 2019*; *Swanson, 1963*). Approximately one in 125 million people worldwide suffer from this rare syndrome. Only a few hundred cases have been described (*Cascella et al., 2022*). To date, about 128 *NTRK1* mutations have been reported in CIPA patients (*Lv et al., 2017*). We have previously reported a series of clinical phenotypes and mutation characteristics of CIPA patients, further expanding our knowledge about the function of the protein encoded by the *NTRK1* gene (*Li et al., 2019*, *2018*; *Wang et al., 2015*, *2016*). However, the neural mechanism behind various symptoms remains elusive. Given the fact that defects in *NTRK1* lead to CIPA, the *NTRK1* is crucial to nociceptive development. On the other hand, the excessive signaling of *NTRK1* results in chronic pain by hypersensitizing pain-mediating neurons (*Hefti et al., 2006*). Remarkably, a growing body of pre-clinical evidence showed that blocking NGF and/or TrkA can alleviate pain and hyperalgesia (*Hsieh et al., 2018*; *Tabata et al., 2012*; *Ugolini et al., 2007*; *Wu et al., 2016*). Clinically, Tanezumab and Fasinumab are the most advanced analgesics targeting the NGF-TrkA signaling pathway in recent years. While, NGF/TrkA inhibitors provided less pain relief than was initially anticipated. Despite a series of setbacks, including side effects and the FDA halting the study at one point, the agency reopened the NGF/TrkA antibody lately. Due to the great value of TrkA/ *NTRK1* for pain treatment, and the dilemma in the new drug development, the specific mechanism of *NTRK1* in neurons needs to be explored urgently.

At present, the specific mechanism of *NTRK1* in neurons are unclear. *NTRK1* is essential for nociceptive sensory and sympathetic neurons to survive, and programmed cell death occurs when neurons lack *NTRK1* (*Levi-Montalcini, 1987*), which hampered studies in depth to a certain degree. In this study, we expressed exogenous *NTRK1* in human SH-SY5Y cells, a neuronal model that does not express TrkA receptors, to study the specific *NTRK1* signal transduction pathway. We obtained a series of differentially expressed genes (DEGs) with *NTRK1* overexpression compared with the control by high-throughput RNA sequencing (RNA-seq). Subsequently, the functionalities of DEGs were predicted using Gene Ontology (GO) and Kyoto Encyclopedia of Genes and Genomes (KEGG) pathway enrichment analyses; then, the hub genes in the protein interaction network (PPI) were

obtained. The results may provide an important clue to elucidate various functions of *NTRK1*-dependent neurons and neurological dysfunction implicated in *NTRK1* including CIPA and chronic pain.

## MATERIALS AND METHODS

### Cloning and plasmid construction

Overexpressed *NTRK1* lentiviruses were purchased from GenePharma Co., Ltd. (Suzhou, China). Transcript information included NM_002529.4, and the lentiviral vector was LV5 (EF-1a/GFP/Puro/Amp).

### Cell culture and transfection

The SH-SY5Y cell lines (Procell Life Science & Technology Co., Ltd., Shanghai, China) were cultured at 37 °C with 5% $CO_2$ in a minimum essential medium (MEM)/F12 (Procell Life Science & Technology Co., Ltd., Wuhan, China) supplemented with 10% fetal bovine serum (FBS; Gibco, New York, NY, USA), 100 µg/mL streptomycin, and 100 U/mL penicillin. SH-SY5Y cells were infected with lentiviruses (multiplicity of infection (MOI) = 35). Infected cells harvested for 48 h were subjected to quantitative reverse transcription-polymerase chain reaction (RT-qPCR), Western blot, and Cell Counting Kit-8 (CCK-8) assays.

### Assessment of gene overexpression

*NTRK1* overexpression was evaluated using Glyceraldehyde-3-phosphate dehydrogenase (GAPDH) as a control gene. In the present study, cDNA synthesis was performed by standard procedures, and RT-qPCR was carried out on the Bio-Rad S1000 with Hieff® qPCR SYBR® Green Master Mix kit (Low Rox Plus; YEASEN, Shanghai, China). As shown in Table S1, the primer sequences were listed. Using the $2^{-\Delta\Delta CT}$ method, the transcript concentration was normalized to the GAPDH mRNA level (*Livak & Schmittgen, 2001*). By using GraphPad Prism, we conducted comparisons *via* the paired Student's t-test (GraphPad Software Inc., San Diego, CA, USA).

### Western blotting

In ice-cold lysis buffer (1× phosphate-buffered saline (PBS), 0.1% sodium dodecyl-sulfate (SDS), 0.5% NP-40, and 0.5% sodium deoxycholate) supplemented with a protease inhibitor cocktail (Roche, Basel, Switzerland), SH-SY5Y cells were lysed and incubated on ice for 30 min. Afterward, we boiled samples with 1× SDS sample buffer for 10 min, separated them by 10% SDS-polyacrylamide gel electrophoresis (SDS-PAGE), and transferred them to membranes. With TBST buffer (20 mM Tris-buffered saline and 0.1% Tween-20) containing 5% non-fat milk powder, membranes were incubated for 1 h at room temperature with FLAG (1:1,000; Cat. No. 2368S; Cell Signaling Technology Inc., Danvers, MA, USA), GAPDH (1:5,000; Cat. No. 60004-1-Ig; Proteintech, Rosemont, IL, USA), GRP78 (1:3,000; Cat. No. 11587-1-AP; Proteintech, Rosemont, IL, USA), p-IRE1 (1:500; Cat. No. AF7150; Affinity, West Bridgford, UK), XBP1s (1:2,000; Cat. No. 24868-1-AP; Proteintech, Rosemont, IL, USA), and ATF6 (1:1,000; Cat. No. DF6009; Affinity, West
Bridgford, UK) primary antibodies, followed by incubation with horseradish peroxidase (HRP)-conjugated secondary antibody. Enhanced chemiluminescence reagent (ECL; Cat. No. 170506; Bio-Rad Laboratories Inc., Hercules, CA, USA) was used to visualize the immunoblots.

## RNA extraction and sequencing

In SH-SY5Y cells, total RNA was extracted with TRIzol reagent (Cat. No. 15596026; Invitrogen, Carlsbad, CA, USA), as described by *Chomczynski & Sacchi (1987)*. After RNA extraction, DNaseI was used to digest DNA. With the Nanodrop™ One Cspectrophotometer system (Thermo Fisher Scientific Inc., Waltham, MA, USA), RNA quality was assessed using A260/A280. Electrophoresis of 1.5% agarose gel was used to confirm RNA integrity. Final quantification of qualified RNAs was performed with the Qubit™ RNA Broad Range Assay kit (Cat. No. Q10210; Life Technologies Corp., Carlsbad, CA, USA) using the Qubit 3.0 Fluorometer.

In addition, KCTM Stranded mRNA Library Prep kit for Illumina (Cat. No. DR08402; Wuhan Seqhealth Co., Ltd., Wuhan, China) was used to prepare stranded RNA sequencing libraries. On the Novaseq 6000 sequencer (Illumina; Chicago, IL, USA) with PE150 model, 200–500 bps PCR products were enriched, quantified, and finally sequenced.

## RNA-seq data processing

First, raw reads with more than 2-N bases were discarded. Then, FASTX-Toolkit (Ver. 0.0.13) was used to trim adaptors and low-quality bases from raw sequencing read. The short reads <16-nt were also dropped. After that, An alignment of clean reads to the GRCh38 genome was performed by HISAT2 (*Kim, Langmead & Salzberg, 2015*) with four mismatches allowed. Reads that were uniquely mapped were counted and fragments per kilobase of transcript per million fragments mapped (FPKM) were calculated (*Trapnell et al., 2010*).

## CCK-8 assay

Cell proliferation was measured using a CCK-8 assay kit (MCE Co., Ltd., Beijing, China). SH-SY5Y cells were seeded into 96-well plates in triplicate at a density of $2 \times 10^4$ cells/well and cultured for 72 h. At 37 °C for 2 h, the cells were then incubated in 10% CCK-8 solution diluted in the MEM/F12. An enzyme-labeling instrument (Bio-Rad Laboratories Inc., Hercules, CA, USA) was used to measure the absorbance (optical density (OD) value) at 450 nm wavelength.

## Analysis of DEGs

DEGs were screened with DESeq2 from R Bioconductor (*Love, Huber & Anders, 2014*). To identify DEGs, we used a *P* value of 0.05 and a fold-change (FC) of 1.5 or 0.67 as cut-off criteria.

## Functional enrichment analysis

With KOBAS 2.0, GO and KEGG pathway enrichment analyses were conducted to sort DEGs into functional categories (*Xie et al., 2011*). To determine the enrichment of each

term, the hypergeometric test and Benjamini-Hochberg false discovery rate (FDR) controlling procedure were applied.

## PPI network construction, modules, and identification of top 10 genes

Search tool for the retrieval of interacting genes (STRING) database (http://stringdb.org/) (*Szklarczyk et al., 2019*) was employed to construct PPI networks by mapping the DEGs into PPI data. The symbols of the DEGs were imported into the database, and high-resolution bitmaps were generated. The bitmap only included interactors with a combined confidence score of 0.4 or higher. The hotspot module was obtained in large PPI networks by the Molecular Complex Detection (MCODE) which is a Cytoscape plugin. In this study, MCODE parameters were as follows: the degree of cut-off = 2; cluster finding, haircut; node score cut-off = 0.2; k-core = 2; and the maximum depth = 100 (*Bader & Hogue, 2003*). The modules with established score >9 and the number of nodes >9 were screened out. The GO enrichment analysis was used to analyze genes in the modules by DAVID. CytoHubba, a plugin for Cytoscape 3.9.1, was used to calculate the degree of connectivity and identify the top 10 genes in the PPI networks (*Shannon et al., 2003*). The PPI networks, modules, as well as the top 10 genes were visualized based on their node degree using Cytoscape software.

## RT-qPCR for validation of the hub genes

An RT-qPCR analysis was performed and normalized with GAPDH to clarify the validity of the top 10 hub genes. RT-qPCR was performed using the same cell lines as RNA-seq. For the amplification program, denaturing was carried out for 30 s at 95 °C, followed by 40 cycles of denaturing at 95 °C for 10 s and 60 °C for 30 s, then annealing and extension at 95 °C for 15 s, 60 °C for 60 s, and 95 °C for 15 s. For each sample, the experiment was conducted in triplicate. Table S1 illustrates the list of the primer sequences used for RT-qPCR analysis.

## Functional annotation of the hub genes

The ClueGO (ver. 2.5.8) and CluePedia (ver. 1.5.8) Cytoscape plugins were performed to decipher the enrichment analysis of the hub genes. GO-biological process (GO-BP) and pathways were considered statistically significant at $P < 0.05$.

## Statistical analysis

According to RNASeqPower, this experimental design has a statistical power of 0.96. The cell biology and RT-qPCR data between the two groups were compared using an unpaired two-tailed t-test. All values were presented as the mean ± standard deviation (SD). Except for the RNA-seq experiment, each experiment was performed at least three times. There was a statistically significant difference at $P < 0.05$.

# RESULTS

## *NTRK1* overexpression promoted the proliferation of SH-SY5Y cells

A lentiviral vector expressing the *NTRK1* gene (*NTRK1*-OE) or an empty lentiviral vector (not expressing the *NTRK1* gene) (NC) was used to transfect SH-SY5Y cells. The results of

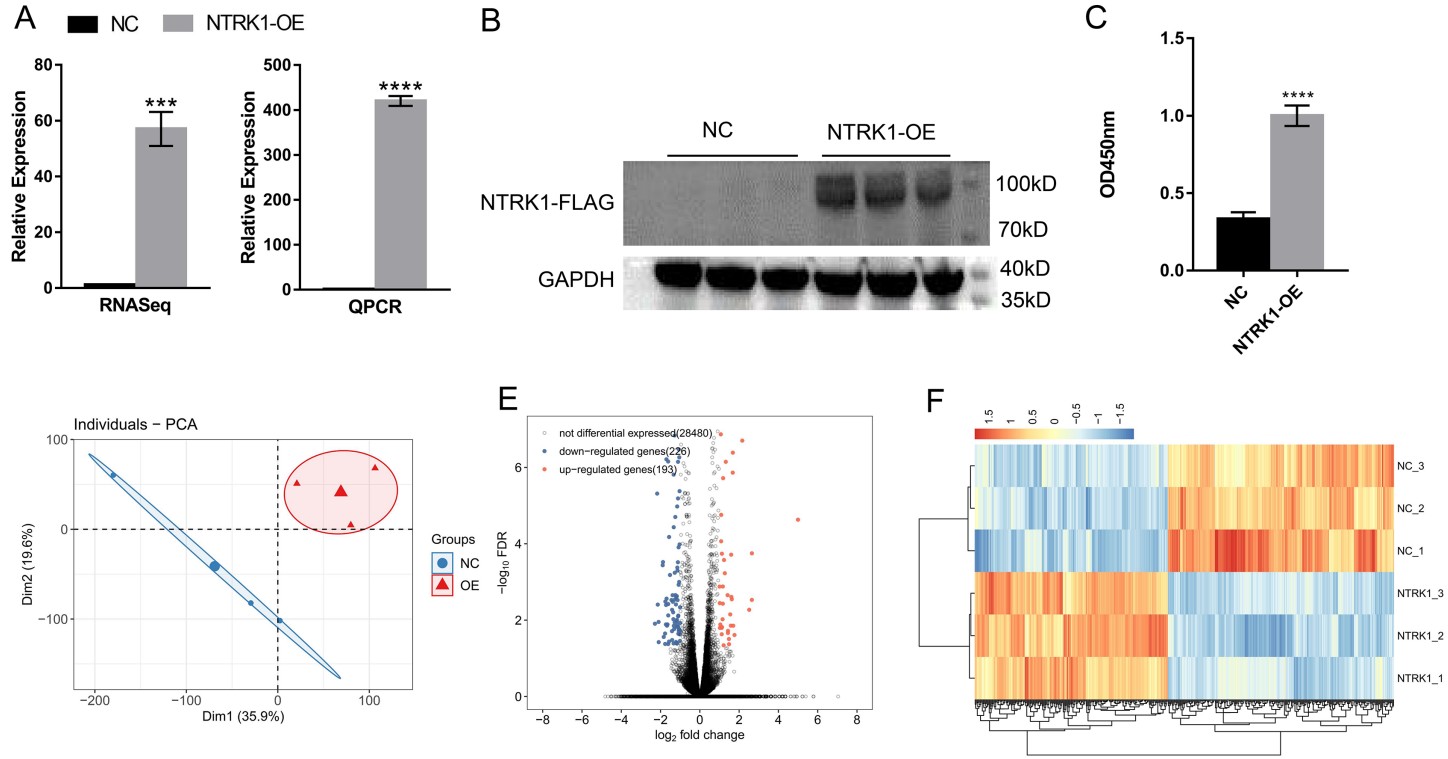

**Figure 1 Analysis of the change in gene expression in response to *NTRK1* overexpression in SH-SY5Y cells.** (A) Bar plot showing the expression levels of *NTRK1* in RNA-seq data and RT-qPCR results. (B) *NTRK1* was overexpressed and verified by Western blotting. Full-length blots/gels are presented in Figs. SB1 and SB2. (C) *NTRK1* overexpression increases the proliferation of SH-SY5Y cells. (D) PCA base on FPKM value of all detected genes. The ellipse for each group is the confidence ellipse. (E) Volcano plot showing all differentially expressed genes (DEGs) between OE and NC samples with DEseq2. FDR < 0.01 and FC (fold change) ≥ 1.5 or ≤ 0.67. (F) Hierarchical clustering heat map showing expression levels of all DEGs. Error bars represent mean ± SEM. ***P-value < 0.001. ****P-value < 0.0001.

the RNAseq, RT-qPCR, and Western blotting showed that *NTRK1* was significantly overexpressed in SH-SY5Y cells (Figs. 1A and 1B). Moreover, the overexpression of *NTRK1* significantly promoted SH-SY5Y cell proliferation (Fig. 1C). The results validated that *NTRK1*-OE was successfully constructed and *NTRK1* favored the survival of the neurons.

## *NTRK1* overexpression altered the gene expression profiles in SH-SY5Y cells

RNA-seq was performed to examine the *NTRK1*-mediated transcriptional regulation in SH-SY5Y cells. In total, six RNA-seq libraries were prepared from *NTRK1*-OE and NC SH-SY5Y cells, with three biological replicates per group (*NTRK1*_1, *NTRK1*_2, and *NTRK1*_3; NC_1, NC_2, and NC_3). After removing low-quality reads and sequence adaptors, a clean pair-end read was obtained for 69.1 million samples on average. Approximately 62.8 million read pairs were uniquely mapped per sample to the human genome (Table S2).

Furthermore, gene expression was calculated using these uniquely mapped reads. A proprietary pipeline was used to calculate FPKM, which represents gene expression levels.

There were 22,154 genes expressed (FPKM > 0), and 10,364 genes expressed at FPKM > 1 in at least one sample (Table S3). A correlation matrix using FPKM values for all 22,154 genes was calculated using Pearson's correlation value (more than 0.99) between *NTRK1*-OE and the control, which indicated the similar expression of most genes. Moreover, *NTRK1*-OE and NC samples were clearly differentiated by principal component analysis (PCA) based on FPKM values for all detected genes (Fig. 1D), and revealed that SH-SY5Y cells overexpressing *NTRK1* showed altered gene expression profiles.

For further analysis of *NTKR1*'s transcriptional regulation, DESeq2 was performed to identify the genes that are differentially expressed between *NTRK1*-OE and NC cells, with a cut-off FC ≥ 1.5 or ≤0.67 and an FDR of 5%. The gene expression level in SH-SY5Y cells was extensively regulated by *NTRK1* overexpression (Fig. 1E), as shown by 193 upregulated and 226 downregulated genes (Table S4). Additionally, the *NTRK1-OE* and NC samples were clearly distinct in the hierarchical clustering analysis of normalized FPKM values of DEGs, and the three replicate datasets also showed a high level of consistency (Fig. 1F). These results indicated that *NTRK1* overexpression significantly changed the transcript expression level of a series of genes in neurons.

## *NTRK1* overexpression affected the biological process in SH-SY5Y cells

The GO and KEGG pathway enrichment analyses of all 419 DEGs were carried out to determine their potential roles. A significant increase in the expression levels of 193 genes was observed. According to the GO enrichment analysis, these genes were mostly enriched in response to endoplasmic reticulum (ER) stress, protein folding in ER, IRE1-mediated unfolded protein response (UPR), collagen fibril organization, ATF6-mediated UPR, cellular protein metabolic process, response to unfolded protein, ER unfolded protein response, folic acid metabolic process, and extracellular matrix (ECM) organization (Fig. 2A). The KEGG pathway analysis further confirmed that the upregulated genes were highly enriched in protein processing in ER (Fig. 2C). Moreover, upregulated genes were also highly enriched in a range of biochemical processes and p53 signaling pathway. A total of 226 genes downregulated by *NTRK1* were mainly enriched in a series of cellular parts and cellular processes, and pathways associated with cell proliferation and migration, such as Dopaminergic synapses, ECM-receptor interactions, transforming growth factor-β (TGF-β) signaling pathway, and Wnt signaling pathway (Figs. 2B and 2D). Overall, these data suggest that a range of biological processes are affected by *NTRK1* overexpression in human neuronal cell lines, in which response to ER stress occupies great importance.

## PPI network analysis and hot modules

A PPI network based on the DEGs was constructed on STRING analysis for functional association and sequentially visualized on Cytoscape, and the results are shown in Fig. 3A. Furthermore, there are 294 nodes and 1,062 edges in the network.

Using the Cytoscape plugin "MCODE", two functional modules (modules 1 and 2) were detected with scores >9 and nodes >9. Module 1 covered 143 edges and 23 nodes, with a

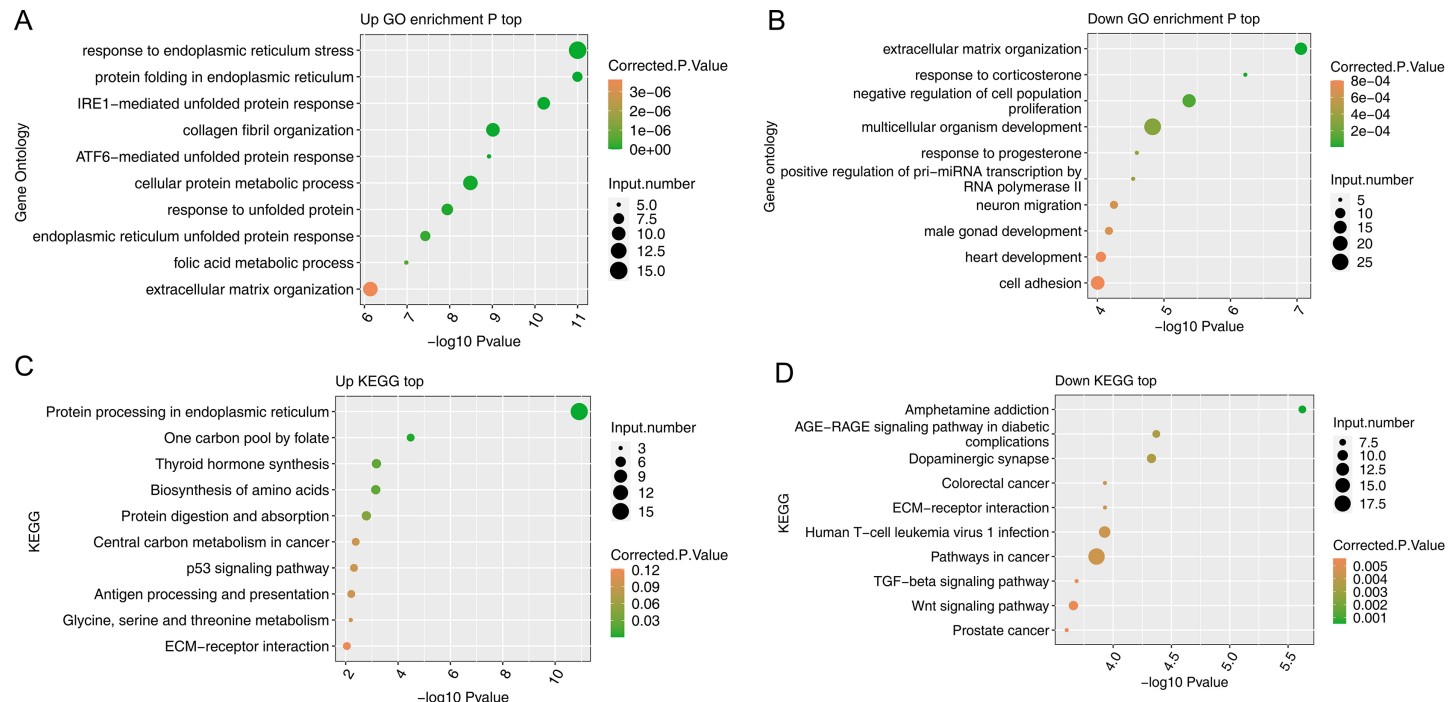

**Figure 2** **Functional analysis of DEGs after *NTRK1* overexpression in SHSY-5Y cells.** (A–D) Bubble diagram exhibiting the most enriched GO biological process and KEGG pathways results of the up or down-regulated DEGs.

score of 13, containing 21 upregulated and two downregulated genes (Fig. 3B). Module 1 functional enrichment analysis revealed that ER stress-related functions dominated the most significant enrichment results in GO-BP (Table 1). Having 42 edges and 10 nodes, module 2 scored 9.33, with 10 genes downregulated (Fig. 3C), and these genes were mainly enriched in mitotic sister chromatid segregation (Table 1). The results of hot modules in the PPI network suggest that ER stress-related functions exert vital roles in neurons with *NTRK1* overexpression.

## Identification of the top 10 genes and validation of hub genes

Using the cytoHubba plugin, the top 10 target genes were identified based on scores, and interactors of the top 10 genes were reconstructed (Fig. 3D). These top 10 genes with detailed information are presented in Table 2.

The expression levels of the top 10 genes were quantified using RT-qPCR to verify the effects of *NTRK1*-OE on them. Overall, seven hub genes were verified, including five upregulated and two downregulated DEGs (Fig. 4A). The upregulated DEGs included COL1A1 ($|log2FC|$ = 0.63), P4HB ($|log2FC|$ = 0.65), HSPA5 ($|log2FC|$ =1.27), THBS1 ($|log2FC|$ = 0.73), and XBP1 ($|log2FC|$ = 0.90); the downregulated DEGs included CCND1 ($|log2FC|$ = 0.73) and COL3A1 ($|log2FC|$ = 1.31). Hub genes tended to be enriched in terms of 'response to endoplasmic reticulum stress', 'cellular response to transforming growth factor beta stimulus', and 'collagen fibril organization' as the biological process

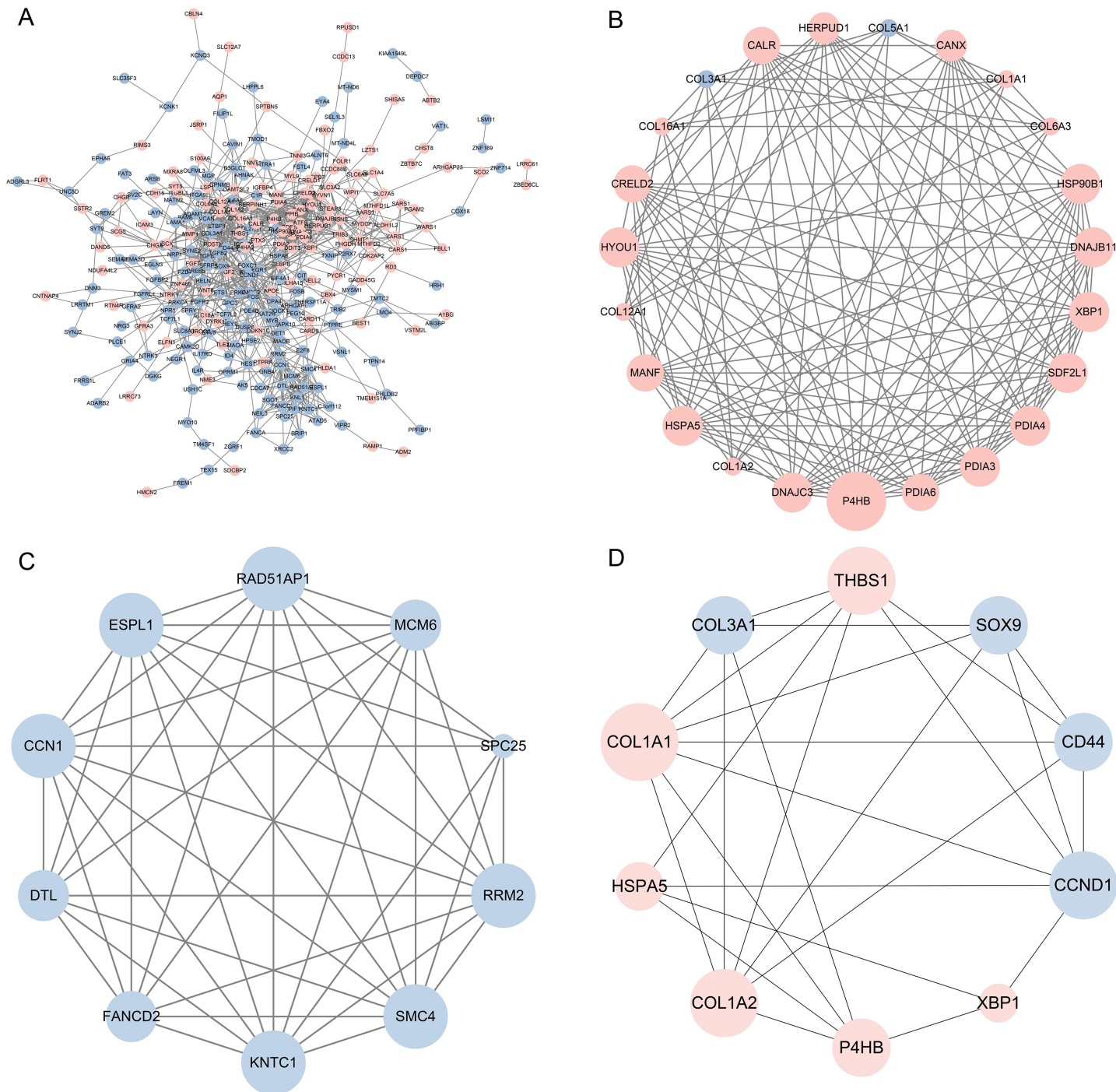

**Figure 3 The PPI network of DEGs were constructed, two significant modules and top 10 genes were identified by Cytoscape.** (A) PPI network of differentially expressed genes. Circles represent genes, and lines represent protein interactions between genes. Data with interaction score >0.4 were selected from protein-protein interactions to construct a PPI network. (B) Module 1, MCODE score = 13. (C) Module 2, MCODE score = 9.33. (D) Top 10 genes of differentially expressed mRNAs. The circle size indicates the degree of association of the gene in the network. Upregulated genes were marked in pink; downregulated genes were marked in blue.

**Table 1  The biological processes associated with modules 1 and 2.**

| Term | P value | Count | Genes |
|---|---|---|---|
| Module 1 | | | |
| Protein folding in endoplasmic reticulum | 3.43E−16 | 7 | DNAJC3, PDIA3, HSPA5, CANX, P4HB, CALR, HSP90B1 |
| Collagen fibril organization | 1.24E−11 | 8 | COL1A1, COL3A1, COL16A1, COL1A2, COL5A1, COL12A1, COL6A3, P4HB |
| Response to endoplasmic reticulum stress | 2.48E−10 | 7 | PDIA3, XBP1, HYOU1, P4HB, HSP90B1, PDIA4, HERPUD1 |
| Protein folding | 6.77E−10 | 8 | PDIA3, CANX, DNAJB11, P4HB, CALR, PDIA6, HSP90B1, PDIA4 |
| IRE1-mediated unfolded protein response | 3.03E−09 | 6 | DNAJC3, XBP1, HSPA5, DNAJB11, HYOU1, PDIA6 |
| ATF6-mediated unfolded protein response | 9.22E−08 | 4 | XBP1, HSPA5, CALR, HSP90B1 |
| Endoplasmic reticulum unfolded protein response | 3.13E−07 | 5 | XBP1, HSPA5, CANX, CALR, HERPUD1 |
| Cellular response to amino acid stimulus | 3.37E−07 | 5 | COL1A1, COL3A1, XBP1, COL16A1, COL1A2 |
| Ubiquitin-dependent ERAD pathway | 1.62E−06 | 5 | HSPA5, CANX, CALR, HSP90B1, HERPUD1 |
| Extracellular matrix organization | 7.19E−06 | 6 | COL1A1, COL3A1, COL16A1, COL1A2, COL5A1, COL6A3 |
| Antigen processing and presentation of peptide antigen *via* MHC class I | 3.87E−04 | 3 | PDIA3, CANX, CALR |
| Cellular protein metabolic process | 5.48E−04 | 4 | DNAJC3, P4HB, PDIA6, HSP90B1 |
| Skin development | 8.97E−04 | 3 | COL1A1, COL3A1, COL5A1 |
| Blood vessel development | 0.001079868 | 3 | COL1A1, COL1A2, COL5A1 |
| Response to unfolded protein | 0.00160925 | 3 | DNAJC3, MANF, HERPUD1 |
| Cell adhesion | 0.002647269 | 5 | COL1A1, COL16A1, COL5A1, COL12A1, COL6A3 |
| Module 2 | | | |
| Mitotic sister chromatid segregation | 1.07E−04 | 3 | ESPL1, KNTC1, SMC4 |

**Table 2  Top 10 genes identified by cytoHubba.**

| Name | Description | Score | Log2 fold change | P value | Regulation |
|---|---|---|---|---|---|
| COL1A1 | Collagen type I alpha 1 chain | 38 | 0.630823936 | 0.000139094 | Up |
| CCND1 | Cyclin D1 | 36 | −0.726736457 | 1.15E−15 | Down |
| P4HB | Prolyl 4-hydroxylase subunit beta | 33 | 0.647655166 | 1.40E−09 | Up |
| HSPA5 | Heat shock protein family A (Hsp70) member 5 | 33 | 1.271269024 | 1.97E−42 | Up |
| THBS1 | Thrombospondin 1 | 30 | 0.733834514 | 0.000724405 | Up |
| COL3A1 | Collagen type III alpha 1 chain | 29 | −1.315967605 | 0.000 | Down |
| SOX9 | SRY-box transcription factor 9 | 28 | −0.862402503 | 1.76E−23 | Down |
| COL1A2 | Collagen type I alpha 2 chain | 27 | 4.199548198 | 1.01E−38 | Up |
| XBP1 | X-box binding protein 1 | 25 | 0.901297817 | 6.24E−18 | Up |
| CD44 | CD44 molecule | 25 | −0.95212 | 2.17E−05 | Down |

(Fig. 4B). Furthermore, the western blot results showed *NTRK1* over-expression increased expression of ER stress markers in neurons, including GRP78, p-IRE, XBP1s, and ATF6 (Fig. S2). Hence, these data further confirm that ER stress response is dominant in neurons with *NTRK1* overexpression.

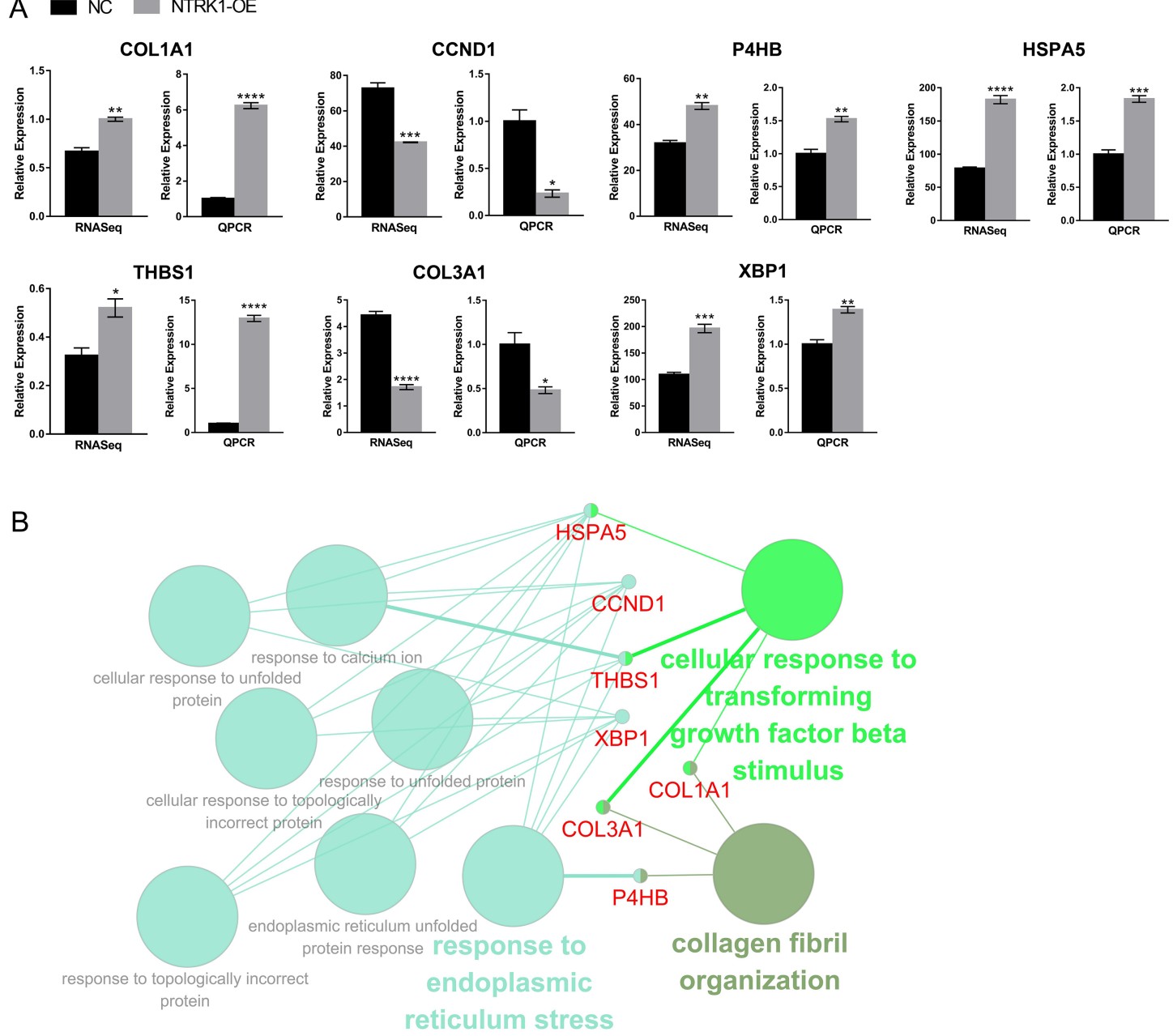

**Figure 4 Identification and functional annotation of hub genes.** (A) Bar plot showing the expression pattern and statistical difference of DEGs. Error bars represent mean ± SEM. $^{*}P$-value < 0.05,$^{**}P$-value < 0.01, $^{***}P$-value < 0.001, $^{****}P$-value < 0.0001. (B) The GO biological processes functional annotation analysis of hub genes. Different colors of nodes refer to the functional annotation of ontologies. The nodes with red font represent the hub genes.

## DISCUSSION

*NTRK1* is a protein-coding gene that encodes TrkA, a member of the neurotrophic tyrosine kinase receptor family. TrkA, a receptor for NGF, transmits NGF-induced signaling cascades necessary for sympathetic and nervous neurons' proliferation, differentiation, and survival. NGF-TrkA contributes to many biological processes in

adults, such as pain, homeostasis, and inflammation (*Indo, 2018*). In the present study, 419 DEGs were obtained in human neuronal cell lines (SH-SY5Y cells) with *NTRK1* overexpression compared with the control by RNA-seq. These DEGs consisted of 193 upregulated and 226 downregulated DEGs. DEGs enriched in GO and KEGG pathways showed a range of biological processes and pathways that were affected by *NTRK1* overexpression, in which response to ER stress occupied great importance. Furthermore, the enrichment analysis of hotspot modules of DEGs revealed that response to ER stress was dramatically enriched in the finest hot modules with MCODE scores >9. Additionally, ER stress was also a key biological process involved in hub genes in the PPI network of DEGs. Taken together, our results demonstrated that *NTRK1* significantly influenced the gene transcription of response to ER stress in neurons.

Previous studies have indicated that the p53 signaling pathway, ECM organization, and TGF-β signaling pathway are affected by *NTRK1*, which is reminiscent of our discovery. It was reported that p53 was regulated through the TrkA and p75 neurotrophin receptors in neonatal sympathetic neurons (*Aloyz et al., 1998*). As for ECM organization in the previous study, the researchers developed a high-throughput screening platform using fibroblast-derived matrices imaged with automated confocal microscopy in 384-well plates to explore mechanisms controlling matrix organization (*Jones et al., 2022*). It was identified that *NTRK1* was one of the modulators of matrix alignment. Moreover, a previous study showed that genetically inhibiting the neural innervation of the TrkA sensory nerve causes premature closure of the calvarial suture with altered TGF-β signaling pathway transcription levels (*Tower et al., 2021*). In our study, in addition to the above pathways, we also found responses to ER stress, protein folding in ER, response to unfolded protein, Dopaminergic synapses, Wnt signaling pathway, and so on were significantly enriched in neurons with *NTRK1* overexpression, especially ER stress response.

The accumulation of misfolded/unfolded proteins that deviates from the norm causes ER stress, which triggers an adaptive response. This response involves activating PERK, IRE1α (inositol-requiring transmembrane kinase/endoribonuclease 1α), and ATF6 (activating transcription factor 6) pathways as part of the UPR/ER stress response (*Hayashi et al., 2007*). The involvement of UPR/ER stress response contributes to neuronal differentiation, neurogenesis, and neurite outgrowth (*Godin et al., 2016*; *Hayashi et al., 2007*; *Hetz & Saxena, 2017*). Thus, we speculated that *NTRK1* may influence various functions of neurons by modulating the expression of genes involved in response to ER stress.

To date, the relationship among ER, UPR, and neuropathology has been well established (*Ghemrawi & Khair, 2020*). The build-up of misfolded proteins within the ER of neurons is a common pathological characteristic of various neurological dysfunctions. In response to ER stress, the UPR minimizes stress, reduces protein misfolding, and maintains cellular homeostasis (*Hetz & Saxena, 2017*). The crosslink between neuroinflammation and ER stress has been recognized in various diseases, including neurodegenerative diseases and pain (*Lei et al., 2021*; *Mao et al., 2020*; *Sprenkle et al., 2017*). Therefore, it can be inferred

that UPR/ER stress response-associated genes may be novel potential targets for neurological dysfunction implicated in *NTRK1*.

In our study, five hub genes implicated in ER stress response were validated, including four upregulated genes (P4HB, HSPA5, THBS1, and XBP1) and one downregulated gene (CCND1). HSPA5 encodes the binding immunoglobulin protein, initiates the UPR, and decreases unfolded/misfolded protein load (*Wang et al., 2017*). XBP1, a key transcription factor, is spliced by IRE1 in response to ER stress to active UPR (*Yoshida et al., 2001*). P4HB is an essential redox-sensitive activator of PERK during the UPR (*Brewer & Diehl, 2000*; *Kranz et al., 2017*), and THBS1 was also shown to activate PERK that mediates the ER stress response (*Vanhoutte et al., 2021*). Moreover, it was reported that CCND1 expression directly targets the UPR (*Bustany et al., 2015*). Therefore, the UPR, especially the IRE1α-XBP1 and PERK signaling pathways, might play key roles in *NTRK1* affecting neurons, which provides insights for the future research directions on the mechanisms of neurological dysfunction implicated in *NTRK1*.

Of greatest significance, however, is the fact *NTRK1* participates in the growth and survival of both sympathetic and sensory neurons, there is very likely a significant limitation in dissecting the exact mechanism of *NTRK1*. To date, few studies have attempted to examine selectively the mechanism of *NTRK1* in neurons. Our study provides more meaningful results in assessing the direct actions of *NTRK1* in neurons. Further, molecular transcriptional responses of *NTRK1* in human neuronal cell lines will offer distinctive opportunities to investigate the crucial roles of *NTRK1* in neurons. Further insight into *NTRK1* signaling may reveal unexpected roles of *NTRK1*-dependent neurons in human physiology, and suggest new options for the treatment of neurological dysfunction implicated in *NTRK1*, including CIPA and chronic pain.

Nevertheless, it should be noted that this study is only taking its first steps, and it has several limitations that must be addressed in future studies. First, our study used SH-SY5Y cells, human neuronal cell lines, rather than native cells. Furthermore, our study does not include animal experiments or research regarding the possible mechanism linking ER stress response to *NTRK1*. Therefore, further studies are needed with animal experiments and the use of native cells to explore the potential causal mechanisms between the ER stress response and *NTRK1* to facilitate future scientific studies and provide guidance for the neurological dysfunction implicated in *NTRK1*.

## CONCLUSIONS

In summary, the present bioinformatics analysis revealed that various ER stress response-related genes were regulated by *NTRK1* overexpression in human neuronal cell lines, although further *in vivo* and *in vitro* validation is required.

## ACKNOWLEDGEMENTS

We are grateful to the team members from ABLife Inc., in guidance and suggestions for the design of the manuscript.

## Funding

The authors received no funding for this work.

## Competing Interests

The authors declare that they have no competing interests.

## Author Contributions

- Bo Jiao performed the experiments, analyzed the data, authored or reviewed drafts of the article, and approved the final draft.
- Mi Zhang performed the experiments, analyzed the data, authored or reviewed drafts of the article, and approved the final draft.
- Caixia Zhang performed the experiments, analyzed the data, prepared figures and/or tables, and approved the final draft.
- Xueqin Cao performed the experiments, analyzed the data, prepared figures and/or tables, and approved the final draft.
- Baowen Liu performed the experiments, prepared figures and/or tables, and approved the final draft.
- Ningbo Li performed the experiments, prepared figures and/or tables, and approved the final draft.
- Jiaoli Sun performed the experiments, prepared figures and/or tables, and approved the final draft.
- Xianwei Zhang conceived and designed the experiments, authored or reviewed drafts of the article, and approved the final draft.

## Data Availability

The data is available at NCBI GEO: GSE221028.

## Supplemental Information

Supplemental information for this article can be found online at http://dx.doi.org/10.7717/peerj.15219#supplemental-information.

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
