# Peer review of "Transcriptomics reveals the effects of NTRK1 on endoplasmic reticulum stress response-associated genes in human neuronal cell lines"

_PeerJ, doi:10.7717/peerj.15219_

## Round 0.1 · original submission · Major Revisions

Dear Authors,

Once again, thank you for submitting your great work with us here at PeerJ. We are pleased to know, read and review your work, and it came up with nice thoughtful comments. It is easy for me to decide on a Major revision, and I don't see you will find difficulties in responding.

The 3 reviewers had done great work and recommended getting a revised and more detailed work from your team. I agree with them, and please find detailed comments on your portal here I highlighted a few.

Review 1 asked what kind of statistical parameters are considered for PPI network analysis and how they have been chosen. Also, please clarify the bias in the analysis part. Also, reviewer 1 mentioned the resolution of figures should be enhanced, especially figure 1. Reviewer 2 also mentioned the same. Please work on it and show us your great work.

Reviewers 1 and 3 also corroborated their observation of the overexpression system of NTRK1 and a more focused introduction. Reviewer 3 mentioned the lack of detailed observation of KEGG and GO. Please clarify.

I am looking forward to seeing the responses and my best wishes.

Regards,
Dr. Nagendran Tharmalingam
Handling Editor- PeerJ

Reviewer 1 ·

Basic reporting

The resolution of each figure should be improved

Experimental design

no comment. Please refer "Validity of the findings" section for details.

Validity of the findings

In this study, the authors have explored how the over-expression of the NTRK1 gene influences ER stress response using RNA-seq data and followed by bioinformatic data analysis. The manuscript could be improved by following directions:

• What is the rationale behind the over-expression of the NTRK1 gene, and the provided information in the manuscript is not sufficient? It needs more explanation.
• In the introduction part, the authors mentioned that many loss-off-function mutations were observed in the CIPA condition. It is curious to know whether these mutations were also observed in the overexpression of NTRK1 samples
• The authors concluded that over-expression of the NTRK1 gene is influenced the ER-stress response. Thus, it is interesting to know is there any experimental validation has been done to support their observations
• From PPI network analysis, how these 2 network modules have been chosen by authors, and what are the statistical parameters considered? It should be explained. Additionally, the distribution/number of up-and-down-regulated genes is not equal in module 1. However, this distribution is opposite to the original number of up (193) and down-regulated (226) genes. Why such biasness has been taken in this analysis part
• The resolution of each figure should be improved. Specifically, figure 1 is not visible completely.
• Figure 2, the title of the x-axis should be changed from “Item” to “Gene ontology” or “biological pathways”. Similarly, what do you mean by “P top” and “F top” in the sentence like “up GO enrichment P top” and “down GO enrichment F top”
• Information on both Up and down-regulated genes, fold change values, p-values, FDR, etc. could be provided as a table.

Reviewer 2 ·

Basic reporting

The introduction, methods, results, and discussion sections are clearly presented.

Experimental design

no comment

Validity of the findings

no comment

Additional comments

1) I request the authors to provide high-resolution images. The current images are too blurry and challenging to interpret.

2) In the introduction section, I request the authors to add the number of CIPA patients globally to know the impact of this disease.

Reviewer 3 ·

Basic reporting

1. The introduction can be improved. Currently the significance of the work is not clear from the introduction. Also the author need to explain how this work in different from the previous work in the field or what gap in knowledge they are trying to address.
2. The authors need to explain the rational for doing this study under overexpression system of NTRK1. Why didn’t they choose a knockdown approach as in the introduction they talk about loss of function mutations on NTRK1 gene.
3. The figure 1 A has both RNA seq data as well as qPCR data. However the text doesn’t talk about RNA seq. Please include this in the text.
4. The authors did the KEGG pathway and GO analysis. They need to comment what is known in the field about the pathways affected by NTRK1 and how is their results similar to dissimilar with the existing work.

Experimental design

1. The gap in knowledge and how this manuscript is addressing them is not clear from this introduction or the discussion.

Validity of the findings

1. The authors also just described the result section. They need to include what they conclude from each result section.
2. The discussion should include how their work will is important and what’s next from here.

---

## Round 0.2 · accepted · Accept

Dear Authors,

Thank you again for submitting your responses that satisfied the peers, and all three peers agreed to accept this work can be published in PeerJ. I appreciate your sincere efforts that resulted in the enhancement of this work that surely qualified for the standards we maintain in PeerJ. The work will next go to the production house, and colleagues in the house will contact you for any needs regarding the proof.

Congratulations on this publication, and looking forward to seeing your next work.

With Kind Regards,

Dr. Nagendran Tharmalingam
Academic Editor
PeerJ Life & Environment

Reviewer 1 ·

Basic reporting

The authors have satisfactorily addressed all of my comments

Experimental design

The authors have satisfactorily addressed all of my comments

Validity of the findings

The authors have satisfactorily addressed all of my comments

Reviewer 2 ·

Basic reporting

All comments have been addressed by the authors.

Experimental design

no comment

Validity of the findings

no comment

Additional comments

no comment

Reviewer 3 ·

Basic reporting

The authors have addressed my comments. I don't have any other comments

Experimental design

The authors have addressed my comments. I don't have any other comments

Validity of the findings

The authors have addressed my comments. I don't have any other comments